# Factors Associated with Depression in Infertile Couples: A Study in Thailand

**DOI:** 10.3390/healthcare11142004

**Published:** 2023-07-12

**Authors:** Tong Yang, Nahathai Wongpakaran, Tinakon Wongpakaran, Ubol Saeng-Anan, Charuk Singhapreecha, Rewadee Jenraumjit, Carmelle Peisah

**Affiliations:** 1Graduate School, Chiang Mai University, Chiang Mai 50200, Thailand; tong_yang@cmu.ac.th (T.Y.); charuk.s@cmu.ac.th (C.S.); rewadee.w@cmu.ac.th (R.J.); carmelle.peisah@health.nsw.gov.au (C.P.); 2Department of Psychiatry, Faculty of Medicine, Chiang Mai University, Chiang Mai 50200, Thailand; 3Division of Reproductive Medicine, Department of Obstetrics and Gynecology, Faculty of Medicine, Chiang Mai University, Chiang Mai 50200, Thailand; ubol.saeng@cmu.ac.th; 4Faculty of Economics, Chiang Mai University, Chiang Mai 50200, Thailand; 5Department of Pharmaceutical Care, Faculty of Pharmacy, Chiang Mai University, Chiang Mai 50200, Thailand; 6Discipline of Psychiatry and Mental Health, Faculty of Medicine, University of New South Wales, Sydney, NSW 2052, Australia; 7Specialty of Psychiatry, Faculty of Medicine and Health, University of Sydney, Sydney, NSW 2006, Australia

**Keywords:** actor–partner interdependence model, depression, couple, dyadic analysis, infertility, mental health

## Abstract

Background: Infertility can affect a couple’s mental health and marital and social relationships. The study aimed to investigate the prevalence of depression among infertile couples and their relationships with other factors. Methods: This study employed a cross-sectional survey. Validated tools were used to assess anxiety and depression, marital satisfaction, personality traits and sufficiency economy. The actor–partner interdependence model (APIM) was used for dyadic analysis. Results: The prevalence of depression in infertile couples was 6.7%. Aggression, extraversion and neuroticism were significantly correlated with depression, whereas the expectation of having children, marital satisfaction and sufficiency economy were negatively correlated with depression. The APIM model suggested that neuroticism and marital satisfaction were significant predictors of depression. Partner effect between the expectation of having children and depression was observed (*p* = 0.039). Conclusions: Like other populations, depression in infertile couples seems to be associated with aggression, extraversion and neuroticism. However, there are specific variables related to infertility that impact the depression levels of these couples. For instance, the expectation of having children can affect the partners of infertile couples, while the role of the sufficiency economy is a new factor that has been examined for depression in this sample and requires further exploration.

## 1. Introduction

According to the World Health Organization (WHO), having sexual intercourse without contraception for more than 12 months without getting pregnant can be considered infertility [1,2]. Male factors, female factors and a mix of factors can all contribute to infertility. However, in the social and cultural context that advocates childbearing, women often bear more blame and pressure. Many families still blame women for infertility. About 20–30% of infertility cases are caused by a combination of male and female factors. Approximately 50% of cases of infertility are due to female factors. Male factors are implicated in at least 30% [3,4,5]. About 85% of infertility cases have a clear cause. That still leaves 15% of infertility unexplained [6].

Infertility can cause stress in couples, producing feelings of sadness, panic, disappointment and so on. During infertility treatment, women may have concerns about body image, miscarriage, childbirth and more [7]. Infertility can affect infertile couples’ mental health, family relationships and social stress [8]. In addition, the prevalence of infertility continues to increase worldwide [9]. Since the 21st century, infertility treatment technology has made great progress. As a result, more and more infertile couples are seeking infertility treatment. Not all infertile couples have a successful outcome, however [10].

As the duration of infertility increases, so does the psychological stress. The effect was particularly pronounced when infertility lasted longer than eight years [11]. Infertility can have negative social and emotional consequences for couples [12]. Depression is one of the main mental problems associated with infertility [13]. The rate of depression among women seeking infertility treatment was very high, at about 17%. Despite advances in assisted reproduction, nearly half of infertile couples never seek infertility treatment because of the high cost and fear of treatment failure [14]. Studies have shown that infertility itself and infertility treatment are associated with poor mental health, particularly depression. The prevalence of depression was higher in infertile couples with lower incomes [15] and those who were unsuccessful in treatment [16]. There is a similar pattern in polygamous areas [15]. Gender is linked to depression in infertile couples. Women have a higher prevalence of depression than men in infertile couples undergoing assisted reproduction, according to a study conducted in Turkey [17]. Among couples who were successfully treated for infertility, those who had twins had a higher level of depression than those who had only one child [13]. The prevention and intervention of infertile couples who are prone to depression is urgent. If not foreseen, it can lead to substance abuse and suicide. And can affect the treatment compliance of infertile couples [16]. It is necessary to pay attention to the mental health of infertile couples and to discover the unknown factors related to depression in infertile couples. This is relevant for the 48 million couples and 186 million individuals worldwide who experienced infertility [1].

Several factors have been found to be related to mental health difficulties among infertile couples, either general factors or specific factors related to infertility. Monthly household income in Thailand has been on the rise since 2004 [18]. Family income affects the mental health of family members. A study in Thailand found that people with lower incomes and poorer economic conditions were at higher risk of mental problems [19]. Therefore, it is meaningful to study income and depression in infertile couples in Thailand.

Couples who had several unsuccessful or no infertility treatments had a lower quality of life than couples who had no infertility history and had at least one child [20]. For patients receiving infertility treatment, the higher the number of infertility treatments, the higher the risk of prenatal depression [21]. Moreover, women who become pregnant after undergoing assisted reproductive technology have a higher risk of depression [22]. In addition, the effect between intimate relationships and mental health goes both ways. Mental problems can affect intimate relationships and vice versa [23]. Having a good marital relationship also has a positive impact on couples’ mental health. The marital relationship is very important to the individual. At the same time, it affects their social life [24].

Personality is one of the determinants of mental health. People with neuroticism tend to have more negative emotions and lower emotional stability and, thus, lower inner strength [25]. In addition, neuroticism is a risk factor for depression [26]. More specifically, depression in infertile women is linked to certain personality traits [27]. More research is needed to explain the relationship between personality traits and depression in infertile couples.

Substance abuse is found to be related to mental health problems. For example, cannabis use has been associated with anxiety, depression and suicide. Cannabis use often precedes depression and suicidal behavior [28]. Persistent use of multiple substances has been linked to the development of depression. It is especially higher in men [29]. People with depression are at increased risk of suicide when they abuse substances [30]. There is a risk of depression with early substance use, such as early use of marijuana, alcohol and other illicit drugs [31]. Still, the relationship between substance use and depression in infertile couples remains unknown.

Moreover, Thai people are influenced by the late King Bhumibol’s sufficiency economy concept. The philosophy of the autarky economy is put forward in the practice of the living standard of Thai people. This principle can be applied to individuals, societies and states [32]. A sufficient economy is enough to support ourselves. This philosophy, based on moderation, reasonableness and immunity, is the way to improve the happiness of life of the Thai people. Moderation means not doing too much and not doing too little. Rationality means making decisions with reason and with care. Immunity refers to the ability to cope with future changes and impacts. A sufficiency economy requires knowledge to help with planning and decision-making, as well as the virtues of honesty, wisdom, patience and perseverance. The sufficiency economy provides a good way to guide people’s lifestyles and social norms. Significantly, following the sufficiency economy can bring people happiness and improve their satisfaction in life [33]. We can infer from this that the sufficiency economy is related to mental health, which may affect depression in infertile couples.

The objective of this study was to investigate the prevalence of depression among infertile couples and the factors associated with it, as mentioned earlier. However, the factors that were examined included both non-specific and specific aspects of infertility and their relevance to the Thai population.

## 2. Materials and Methods

The study was designed to explore the relationship between sociodemographic factors and personal history, personality traits, marital satisfaction, sufficiency economy and depression in infertile couples while the interaction between partners. Actor–partner interdependence model (APIM) was a dyad model used to study the interaction between couples. It meant that in couples, depression was not only affected by the self but also by the spouse. More details can be found in Figure 1.

Scope of the study

A cross-sectional study was conducted. The population was infertile couples who attended CMEx Fertility Center, under the Faculty of Medicine, Chiang Mai University and Chiang Mai IVF (In Vitro Fertilization) Polyclinic. Convenience sampling was carried out. This phase was conducted on random days and in random centers. The study was run at CMEx Fertility Center under the Faculty of Medicine, Chiang Mai University and Chiang Mai IVF Polyclinic during July–August 2022.

Participants

A total of 150 infertile couples (300 participants) enrolled in the study at CMEx Fertility Center under the Faculty of Medicine, Chiang Mai University and Chiang Mai IVF Polyclinic.

Inclusion criteria

(1)At least one spouse has been diagnosed with infertility and has consulted at the CMEx Fertility Center under the Faculty of Medicine, Chiang Mai University or Chiang Mai IVF Polyclinic;(2)The couples could read and write Thai or English;(3)Both spouses agreed to participate in the study.

Exclusion criteria

(1)One spouse disagreed to participate;(2)The physician determined that the patient’s vital signs were unstable or there was a medical emergency.

Sampling

In this study, random sampling was used to reduce selection bias [34]. The student PI and the research assistant invited participants to answer the questionnaire at CMEx Fertility Center, under the Faculty of Medicine, Chiang Mai University, on Mondays, Wednesdays and Fridays. The research team invited participants to answer the questionnaire at Chiang Mai IVF Polyclinic on Tuesdays, Thursdays and Weekends. 

The study was approved by the Research Ethics Committee of Faculty of Medicine, Chiang Mai University. Data were collected under the supervision of the visiting physician at both centers. Participants could choose to fill out the paper version of the questionnaire or the digital version. Each member of the couple had to fill out a questionnaire separately. A questionnaire filled out by only one person is considered invalid. Data collection ended after sufficient sample size was obtained. In Figure 2, out of a total of 192 couples, 150 couples were finally recruited into this study.

### 2.1. Measurements

The measurements were available in Thai and English. In addition, considering the situation of COVID-19, participants could choose to fill out a paper or digital version. The measurements included are as follows:Sociodemographic information: The demographic information packet required basic sociodemographic information, including gender, age, education, monthly income, the expectation of having children and the number of infertility treatments.Outcome Inventory-21 (OI-21): OI-21 was created in 2022 by Wongpakaran et al. which was a self-rating questionnaire used to measure levels of depression, for which internal consistency, test–retest reliability, convergent and discriminant validity and diagnostic performance have been confirmed. It was a self-rating questionnaire used to measure levels of depression. Anxiety, somatization and interpersonal difficulties were also measured. It was a five-point scale, ranging from 0 (Never) to 4 (Almost Always). There were 21 questions in total [35].Zuckerman–Kuhlman–Aluja Personality Questionnaire (ZKA-PQ): The original version of ZKA-PQ was created in 2010 by Aluja et al., including 5 personality traits. The short version had 40 questions. It was a four-point questionnaire, from 1 (Strongly Disagree) to 4 (Strongly Agree). The five-factor structure includes neuroticism (NE), sensation seeking (SS), extraversion (EX), activity (AC) and aggressiveness (AG) [36].ENRICH (evaluation and nurturing relationship issues, communication and happiness) marital satisfaction scale was developed in 1993 by Fowers et al. to assess marital satisfaction. It had 15 questions, and 5 of them were negative ratings. ENRICH scale included communication, resolving family conflicts, family roles, financial problems, free time, sexual relationships, child rearing, family and friends and religion [37].Sufficiency economy scale (SES) was created in 2022 by Wongpakaran to determine the level of sufficiency economy. There were 9 questions, on a seven-point scale, from 1 (Strongly Disagree) to 7 (Strongly Agree). The better the understanding and practice of the sufficiency economy, the higher the score [38].

The Cronbach’s alpha of OI-21, ZKA-PQ, ENRICH scale and SES were 0.937, 0.753, 0.930 and 0.750, respectively. 

### 2.2. Data Analysis

Descriptive analyses were applied to sociodemographic data and scores of mental health outcomes, mainly in terms of frequency, percentage, mean and standard deviation.

For the difference test, the t-test and the χ^2^ test were, respectively, applied according to the continuous data and the categorical data, e.g., educational level and substance use. ANOVA was used to test the differences in depression among multiple groups, e.g., the total score of depression among different groups of occupations. The correlation between variables and depression scores was checked by Pearson’s correlation and point-biserial correlation. Multiple regression was used to analyze the predictors of depression. Interaction term analyses were performed to investigate potential interactions among various subgroups (e.g., based on sex). Furthermore, variables suspected to have a significant interacting effect (e.g., predictors demonstrating substantial effect sizes) were examined. If the interaction terms produced significant results, they were subsequently included in the model.

The analysis of APIM was carried out by the multilevel modeling written by Kenny [39]. The coefficient analysis employed t-tests and Z-tests. Using the standard deviations of all at once and the standard deviations of a single parent at once. Values above *r* = 0.10 were small effect sizes. Between *r* = 0.30 and *r* = 0.50 was a moderate effect size. Above *r* = 0.50 was a large effect size [40]. SPSS version 22 was used for data analysis. The results were statistically significant when the *p*-value was less than 0.05 with a 95% confidence interval.

## 3. Results

### 3.1. Sociodemographic Characteristics of the Participants

A total of 300 participants (150 couples) were included in the study. The sociodemographic characteristics of the participants are shown in Table 1. The mean age of men was slightly higher than that of women, *t* (280) = 2.824, *p* < 0.01. More than a third of the participants were self-employed, or about 35.2%. 

More than half of the participants had a bachelor’s degree, accounting for 58.7%. The majority of monthly income in participants was less than or equal to 760 USD. Most participants strongly agreed with the expectation of having children, at about 78.0%. The majority of the participants (46.0%) had received one infertility treatment. More than 90% of the participants did not smoke. More than half of the participants did not drink alcohol. 

Regarding the prevalence of depression, it was 6.7%, equally in males and females. No significant gender differences were observed in other sociodemographic characteristics between males and females.

### 3.2. Descriptive Statistics and Test Difference

Females had significantly higher mean scores of neuroticism than males. The remaining results showed no statistically significant differences in gender (Table 2).

### 3.3. Test Differences between Sociodemographic Factors and Depression

Sociodemographic factors and depression were not statistically significant among participants (Table 3), but occupation (female only, *F* (4) = 4.223, *p* < 0.01).

### 3.4. Pearson’s Correlation between Variables and Depression

Pearson’s correlation showed that aggression and neuroticism were positively correlated with depression, while the expectation of having children, ENRICH marital satisfaction and sufficiency economy were negatively correlated with depression.

As with all participants, aggression and neuroticism were positively correlated with depression in male participants. ENRICH marital satisfaction was negatively correlated with depression in male participants.

Similarly, aggression and neuroticism were positively correlated with depression in female participants. Alcohol and substance use were positively correlated with depression in female participants. The expectation of having children, ENRICH marital satisfaction and sufficiency economy were negatively correlated with depression in female participants. More details are shown in Table 4.

### 3.5. The Multiple Regression Predicting Depression Symptom

Regression analysis was performed for the variables correlated with depression. The results showed that neuroticism (*p* = 0.001) and marital satisfaction (*p* = 0.015) were predictors of depression in all participants. Marital satisfaction was a predictor of depression in male participants, *p* = 0.030. Neuroticism was a predictor of depression in female participants, *p* = 0.001 (Table 5).

### 3.6. The Effect of Variables on the Depression of the Partner on APIM

Instead of studying fathers and mothers separately, the authors examined the interactions and dynamics between fathers and mothers as a unit or partnership in dyad analysis to understand how the two parents collaborated, communicated and jointly influenced family processes and outcomes. The focus of this study was the investigation of the effect of the expectation of having children, AG, NE, ENRICH and SES on depression. Both the effect of own expectation, AG, NE, ENRICH and SES (actor) and the effect of partner’s expectation, AG, NE, ENRICH and SES (partner) on depression was studied. Separate actor and partner effects were estimated for husbands (males) and wives (females), the dyad members being distinguishable by their sex. For the APIM analysis, there were a total of 148 dyads and 296 individuals, with 4 individuals dropped from the analysis due to missing data on one or more variables, with a total of 148 husbands and 148 wives. No other independent variables predicted depression, except for the expectation of having children and neuroticism. No interaction terms were significant for each model; therefore, they were not included in each model.

Regarding the expectation of having children, only the combined partner effect across both husbands and wives was significant (B= −0.323, *p* = 0.039). The standardized effect equaled −0.083 (r = −0.126), which was considered a small effect size (Figure 3).

Regarding the variable of neuroticism, the combined actor effect across both husbands and wives equaled 0.092 and was statistically significant (*p* = 0.003), and the standardized effect equaled 0.166 (*r* = 0.182 and a small effect size) (Figure 4).

Regarding the variable of marital satisfaction, the combined actor effect across both husbands and wives equaled −0.022 and was statistically significant (*p* = 0.039). The standardized effect equaled −0.084 (*r* = −0.126), which was considered a small effect size (Figure 5).

## 4. Discussion

The aim of this study was to determine the prevalence of depression and its associated factors in Thai infertile couples. In general, major depression symptoms were observed among infertile couples, even though it was not high. Unlike the other related study in India, in which the prevalence of depression was 58%, the current study revealed a lower rate of depression, at 6.7%. This may be attributed to the fact that the measurements. The present study used an outcome inventory and a self-report questionnaire, whereas the survey in India used the Hamilton depression rating scale, a clinician-rated measurement [41]. The measurement used by clinicians tends to detect the high rate of major depression rather than a self-reporting questionnaire [42].

Traditionally, female factors have been considered the main reason for infertility assessment [43]. Among infertile couples, women suffer more negative consequences than men, including physical problems, stress and discrimination [44]. In line with the other related study, the current results indicated that women in couples undergoing infertility treatment had significantly higher stress, anxiety and depression scores than men [45]. The same is true for neuroticism; our findings showed females have a higher level of neuroticism personality trait than males, and neuroticism was a strong predictor for depression [46].

Interestingly, the expectation of having children was associated with decreased depression. It seems to be a positive rather than a negative variable. A related study showed that not having children is a risk factor for anxiety in women [47]. The higher the level of depression in infertile women, the less likely they were to seek infertility treatment. If infertile couples have higher expectations of having children, they have more positive attitudes toward infertility treatment [48]. We hypothesized that the fact that the expectation of having children was considered hope rather than stress because almost all participants were in the early phase of treatment. If the study is longitudinal, the expectation might become a stressor, especially for those who fail the treatment. Significantly, societal norms regarding the necessity of couples having children may vary across different cultures [49]. However, irrespective of these cultural differences, the presence of support from one’s partner remains crucial. An insightful study conducted in China highlighted that couples undergoing in vitro fertilization (IVF) expressed mixed feelings when it came to receiving social support from their family members during the process [50]. This evidence is indirectly demonstrated in the current study through the correlation between marital satisfaction and depression.

Consistent with our hypothesis, we observed a negative association between the sufficiency theory concept and depression. This finding aligns with the notion that social values play a significant role in shaping individuals’ attitudes toward life, particularly in the face of challenging and disheartening experiences. It is worth noting that this variable represents a positive attitude specific to Thai culture. However, it is recommended to conduct replication studies in different cultural contexts to ensure generalizability.

Finally, neuroticism appeared to be the most potent factor for depression, and it has been widely recognized as a strong predictor of depression. This is to indicate that the intra-personal factor (personality trait-neuroticism) is more powerful than extra-personal factors.

In dyadic analysis, only three variables were predictors for depression. The actor effects were observed in marital satisfaction and neuroticism, whereas the partner effect (not actor effect) of having children was observed as a predictor for individual depression.

While it is evident that there are gender differences in the negative effects of infertility, it cannot be denied that infertility places a significant burden on the marital life of couples [44]. It is anticipated that marital satisfaction and neuroticism demonstrate to have an actor effect on depression endorsed by the related research [47]. However, this is the first to report that the expectation (hope) of having children demonstrated the partner effect.

### 4.1. Clinical Implication

Research has shown that psychological interventions for women seeking infertility treatment can help improve their psychological problems [51]. From the present findings, some knowledge can be applied clinically. Expectations of having children should be cultivated, especially in the early phase of treatment, as it may be associated with a low incidence of depression. The personality trait of neuroticism, the strongest predictor, can be screened to prepare individuals to cope with stress or anxiety, especially when treatments are unexpectedly disappointing. In the process of infertility treatment, clinicians need to understand and screen the risk factors related to the mental health of infertile couples [52]. It is helpful to provide psychoeducation to patients, e.g., classes info sessions, pamphlets in the waiting room and so on.

### 4.2. Limitations of the Study

Some limitation of our research was that we invited infertile couples who sought treatment. Response and recall biases might occur because the data were collected with a self-administered questionnaire. The researchers considered social desirability bias that could occur as the participants were couples. The fact that we have not included questions about polygamy, religion and whether the couples have extramarital relationships. The cross-sectional nature of the research limited causal relationships of the outcome.

## 5. Conclusions

This study provided evidence of the prevalence of depression and the associated factors with depression in infertile couples. Intrapersonal factors such as neuroticism remain the strongest predictor for depression. Other related factors with infertility, i.e., marital satisfaction and expectation of having children, cannot be overlooked. Outcomes will raise awareness about mental health problems among infertile couples and guide future research for interventions; findings from our study benefit clinicians in identifying a case with such risk factors for anxiety and depression.

## Figures and Tables

**Figure 1 healthcare-11-02004-f001:**
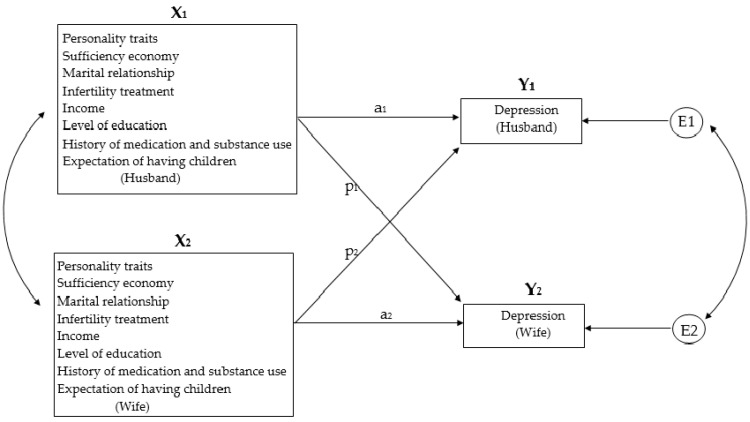
APIM shows the actor effect, partner effect and interaction effect in infertile couples, e.g., X_1_: marital satisfaction; X_2_: wives’ marital satisfaction; Y_1_: husbands’ depression; Y_2_: wives’ depression; E1 and E2: corresponding error terms.

**Figure 2 healthcare-11-02004-f002:**
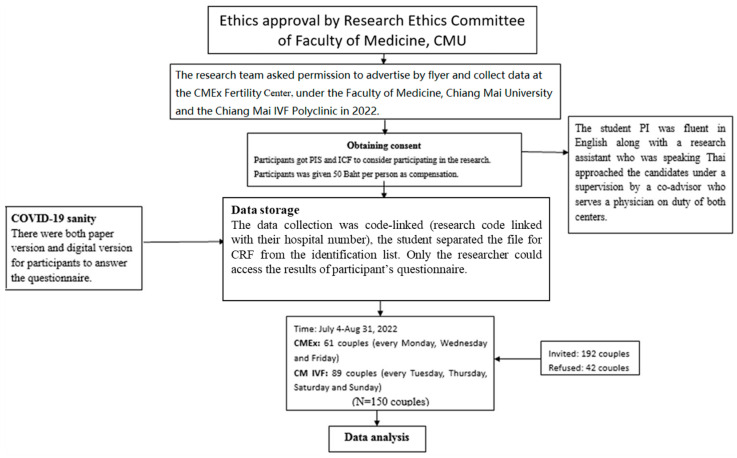
Flow diagram of the study.

**Figure 3 healthcare-11-02004-f003:**
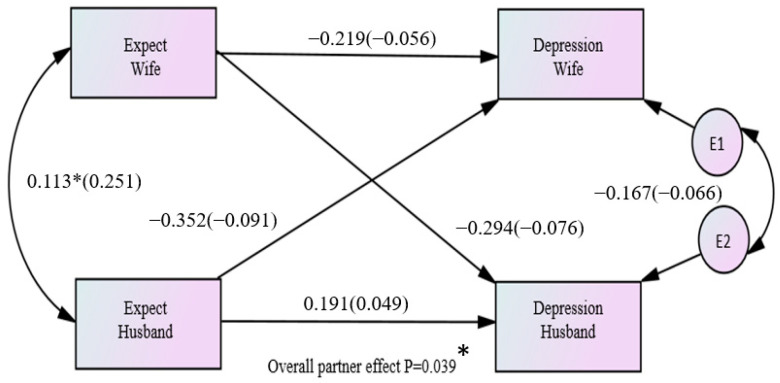
Overall effects between the expectation of having children and depression among couples (* *p* < 0.05).

**Figure 4 healthcare-11-02004-f004:**
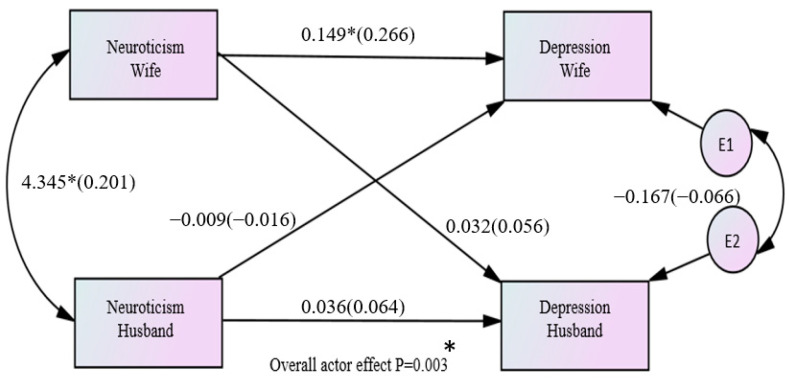
Overall effects between neuroticism and depression among couples (* *p* < 0.05).

**Figure 5 healthcare-11-02004-f005:**
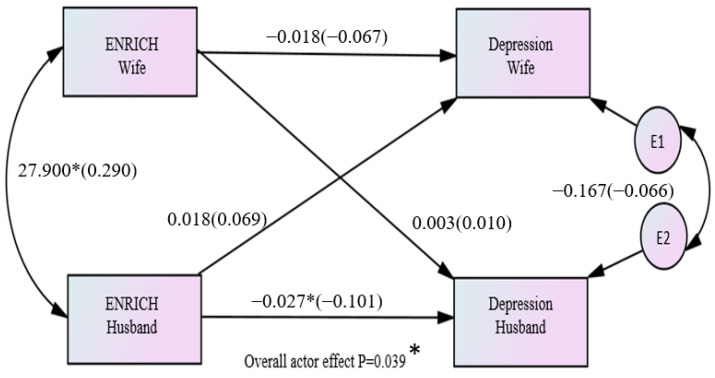
Overall effects between marital satisfaction and depression among couples (* *p* < 0.05).

**Table 1 healthcare-11-02004-t001:** Sociodemographic characteristics of the participants.

Variables		N (%) Mean ± SD	N (%) Mean ± SD	N (%) Mean ± SD	Test Difference
		Male (N = 150)	Female (N = 150)	Total (N = 300)	
Age		36.55 ± 5.98 (20–62)	34.81 ± 4.61 (21–51)	35.68 ± 5.40 (20–62)	*t* (280) = 2.824, *p* < 0.01
Occupation	Freelance	22 (7.4%)	20 (6.7%)	42 (14.1%)	χ^2^ (4) = 3.029, *p* = 0.553
	Government or state enterprise	46 (15.4%)	46 (15.4%)	92 (30.9%)	
	Self-employed	56 (18.8%)	49 (16.4%)	105 (35.2%)	
	Unemployed	1 (0.3%)	4 (1.3%)	5 (1.7%)	
	Other	24 (8.1%)	30 (10.1%)	54 (18.1%)	
Education	Illiterate	0	0	0	χ^2^ (4) = 2.071, *p* = 0.723
	Primary school	1 (0.3%)	3 (1.0%)	4 (1.3%)	
	High school	19 (6.3%)	16 (5.3%)	35 (11.7%)	
	Vocational school	23 (7.7%)	18 (6.0%)	41 (13.7%)	
	Bachelor’s degree	85 (28.3%)	91 (30.3%)	176 (58.7%)	
	Higher	22 (7.3%)	22 (7.3%)	44 (14.7%)	
Monthly Income	0–25,000	64 (21.3%)	74 (24.7%)	138 (46.0%)	χ^2^ (4) = 3.410, *p* = 0.492
	25,001–50,000	55 (18.3%)	56 (18.7%)	111(37.0%)	
	50,001–75,000	14 (4.7%)	8 (2.7%)	22 (7.3%)	
	75,001–100,000	7 (2.3%)	4 (1.3%)	11 (3.7%)	
	100,001 or higher	10 (3.3%)	8 (2.7%)	18 (6.0%)	
Expect to have children	Strongly disagree	0	3 (1.0%)	3 (1.0%)	*t* (298) = 0.342, *p* = 0.732
	Moderately disagree	0	1 (0.3%)	1 (0.3%)	
	Neither agree nor disagree	13 (4.3%)	2 (0.7%)	15 (5.0%)	
	Moderately agree	18 (6.0%)	29 (9.7%)	47 (15.7%)	
	Strongly agree	119 (39.7%)	115 (38.3%)	234 (78.0%)	
Infertility treatment times	1	72 (24.0%)	66 (22.0%)	138 (46.0%)	*t* (298) = −1.126, *p* = 0.261
	2	31 (10.3%)	39 (13.0%)	70 (23.3%)	
	3	13 (4.3%)	12 (4.0%)	25 (8.3%)	
	4	4 (1.3%)	5 (1.7%)	9 (3.0%)	
	5	2 (0.7%)	2 (0.7%)	4 (1.3%)	
	6	2 (0.7%)	3 (1.0%)	5 (1.7%)	
	7	0	1 (0.3%)	1 (0.3%)	
Smoke	No	123 (41.0%)	150 (50%)	273 (91.0%)	χ^2^ (1) = 29.670, *p* (Fisher’s) < 0.001
	Yes	27 (9.0%)	0	27 (9.0%)	
Alcohol	No	48 (16.0%)	111 (37.0%)	159 (53.0%)	χ^2^ (1) = 53.11, *p* < 0.001
	Yes	102 (34.0%)	39 (13.0%)	141 (47.0%)	
Other substance use	No	48 (16.0%)	111 (37.0%)	159 (53.0%)	χ^2^ (1) = 53.111, *p* < 0.001
	Yes	102 (34.0%)	39 (13.0%)	141 (47.0%)	
Infertile relatives	No	133 (44.3%)	126 (42.0%)	259 (86.3%)	χ^2^ (1) = 1.384, *p* = 0.313
	Yes	17 (5.7%)	24 (8.0%)	41 (13.7%)	
Prevalence of depression		10 (6.7%)	10 (6.7%)	20 (6.7%)	χ^2^ (1) = 29.670, *p* (Fisher’s) = 1.000

**Table 2 healthcare-11-02004-t002:** Test difference and descriptive statistics.

Variables	N (%), Mean ± SD			Test Difference
	Male (N = 150)	Female (N = 150)	Total (N = 300)	
Depression	1.67 ± 2.451	1.93 ± 2.735	1.80 ± 2.60	*t* (298) = −0.845, *p* = 0.399
AG	15.07 ± 4.370	14.95 ± 4.659	15.01 ± 4.51	*t* (298) = 0.230, *p* = 0.818
SS	24.86 ± 3.698	25.41 ± 4.241	25.14 ± 3.98	*t* (298) = −1.204, *p* = 0.229
AC	20.50 ± 3.417	20.27 ± 3.775	20.39 ± 3.60	*t* (298) = 0.545, *p* = 0.586
EX	21.67 ± 2.298	21.09 ± 3.011	21.38 ± 2.690	*t* (298) = 1.875, *p* = 0.062
NE	14.61 ± 4.356	15.87 ± 4.890	15.24 ± 4.67	*t* (298) = −2.356, *p* < 0.05
ENRICH	52.85 ± 9.781	53.81 ± 9.858	53.33 ± 9.82	*t* (298) = −0.847 *p* = 0.398
SES	35.53 ± 6.280	36.12 ± 8.912	35.82 ± 7.70	*t* (268) = −0.667, *p* = 0.506

(AG = aggression, SS = sensation seeking, AC = activity, EX = extraversion and NE = neuroticism, refer to ZKA-PQ. ENRICH = ENRICH marital satisfaction scale, SES = sufficiency economy scale).

**Table 3 healthcare-11-02004-t003:** Test differences between sociodemographic factors and depression in all participants (N = 300).

Variables	Test Differences
Occupation	*F* (4, 293) = 2.795, *p* < 0.05
Education	*F* (4, 295) = 0.387, *p* = 0.818
Monthly Income	*F* (4, 295) = 0.362, *p* = 0.836
Smoke	*t* (298) = −0.109, *p* = 0.914
Alcohol	*t* (298) = −0.811, *p* = 0.418
Other substance use	*t* (298) = −0.811, *p* = 0.418
Infertile relatives	*t* (298) = −0.401, *p* = 0.689

*t* = t-statistic, *F* = F-statistic.

**Table 4 healthcare-11-02004-t004:** Correlations between variables and depression.

	All Participants (N = 300)	Male Participants (N = 150)	Female Participants (N = 150)
Age	−0.037	−0.023	−0.038
Gender	0.049		
Expectation	−0.121 *	0.020	−0.228 **
Smoke	0.006	0.034	
Alcohol	0.047	−0.033	0.178 *
Other substance use	0.047	−0.033	0.178 *
Infertile relatives	0.023	−0.012	0.045
Infertility treatment times	−0.019	0.033	−0.066
AG	0.317 **	0.294 **	0.339 **
SS	−0.066	−0.085	−0.058
AC	0.099	0.055	0.137
EX	0.133 *	0.055	0.201 *
NE	0.601 **	0.546 **	0.644 **
ENRICH	−0.209 **	−0.225 **	−0.201 *
SES	−0.157 **	−0.214 **	−0.080

* *p* < 0.05, ** *p* < 0.01, AG = aggression, SS = sensation seeking, AC = activity, EX = extraversion and NE = neuroticism, refer to ZKA-PQ. ENRICH = ENRICH marital satisfaction scale, SES = sufficiency scale, aggression, extraversion and neuroticism refer to ZKA personality traits.

**Table 5 healthcare-11-02004-t005:** Multiple regression predicting anxiety and depression symptoms.

Variables	B	Standard Error	*β*	*p*-Value
Whole sample (N = 300)				
Aggression	−0.005	0.025	−0.008	0.849
Extraversion	0.005	0.036	0.005	0.897
Neuroticism	0.089	0.028	0.160	0.001
Sufficiency economy	−0.008	0.022	−0.014	0.720
Expectation	−0.150	0.138	−0.039	0.279
Marital satisfaction	−0.025	0.010	−0.093	0.015
Male (N = 150)				
Aggression	0.006	0.035	0.011	0.863
Extraversion	−0.015	0.057	−0.014	0.789
Neuroticism	0.035	0.042	0.063	0.403
Sufficiency economy	−0.018	0.034	−0.031	0.595
Expectation	0.101	0.212	0.025	0.635
Marital satisfaction	−0.031	0.014	−0.122	0.030
Female (N = 150)				
Aggression	−0.016	0.035	−0.027	0.649
Extraversion	−0.007	0.048	−0.008	0.887
Neuroticism	0.131	0.039	0.233	0.001
Sufficiency economy	−0.015	0.030	−0.027	0.631
Expectation	−0.326	0.195	−0.087	0.096
Marital satisfaction	−0.012	0.015	−0.042	0.436

## Data Availability

Data not applicable.

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
