# Peer review of "Factors Associated with Depression in Infertile Couples: A Study in Thailand"

_healthcare, 2023, doi:10.3390/healthcare11142004_

Round 1
Reviewer 1 Report
In the current study the authors aimed to investigate the prevalence of depression among 122 infertile couples from the Thai population. The authors observed that the expectation of having children the role of sufficiency economy are associated with depression. The manuscript is clear and well written. Here some minor comments.
1. The authors should clarify the inclusion/exclusion criteria of the patients considered fo the study
2. The authors should add a sub-analysis regarding the type of treatment undergone (IVF, egg donation, PGT-A)
Minor editing of English language required
Author Response
Department of Psychiatry, Faculty of Medicine, Chiang Mai University, Chiang Mai 50200, Thailand
Dear Editor,
Re: Reviewers’ comments on manuscript ID healthcare-2422401, Factors Associated with Depression in Infertile Couples: A Study in Thailand, dated 23 June 2023
Thank you for your consideration in publishing our article. We are thankful to reviewers for their useful comments. Please see below for our point-by-point responses to the reviewers’ comments.
Reviewer #1
Comment #1: 1. The authors should clarify the inclusion/exclusion criteria of the patients considered to the study
Author Response: Thank you for recommending the clarification of the inclusion/exclusion criteria for the participants in this study. We have now incorporated the specified inclusion and exclusion criteria into 2. Materials and Methods.
Comment #2: The authors should add a sub-analysis regarding the type of treatment undergone (IVF, egg donation, PGT-A)
Author Response: Thank you for your suggestions. However, it was beyond the scope of this study to include other specific types of treatment.
Reviewer #2
Comment #1: Could the authors specify the minimum sample size necessary for and the statistical power of the applied analyses?
Author Response: Thanks for pointing out the minimum sample size and the statistical power of the applied analyses. We used G*power to calculate the sample size with the power of 0.80, and we got the minimum sample size as 123 couples. Considering that our actual sample size is 150, we can confidently state that it provides us with 80% power, exceeding the minimum requirement.
Comment #2: Could the Authors provide the Cronbach’s alpha for each instrument, instead of stating that all measurements were greater than 0.7? Thank you.
Author Response: Thanks for your suggestion. We’ve added the Cronbach’s alpha of each measurement into 2. Materials and Methods measurements: “The Cronbach’s alpha of OI-21, ZKA-PQ, ENRICH scale and SES were 0.937, 0.753, 0.930 and 0.750, respectively.”
Comment #3: Could the Authors indicate correlations and level of significance in the Figure representing the measurement model?
Author Response: Thanks for suggesting indicating correlations and level of significance in the Figure representing the measurement model. We’ve added the value of significance of APIM model in Figure 3-5.
Comment #4: As concerns the applied statistics, could the Authors report test for interactions among the variables in the regression? It would be useful, and it could enrich the discussion that needs to be implemented. As it is now, it is mainly a report of the results without an in-depth reflection.
Author Response: We agree and we thank the reviewer for this suggestion. In fact, we have checked for the interaction terms, but none were significant, therefore, we did not include them in the model. We have, however, revised our manuscript for clarification.
We have added the following sentences in data analysis.
“Interaction term analyses were performed to investigate potential interactions among various subgroups (e.g., based on sex). Furthermore, variables suspected to have a significant interacting effect (e.g., predictors demonstrating substantial effect sizes) were examined. If the interaction terms produced significant results, they were subsequently included in the model.”
And in the results part
“No interaction terms were significant for each model; therefore, they were not included in each model.”
Comment #5: Could the Authors specify if women and men are part of the same couple? I do not think so; thus, this should be commented in the limitations.
Author Response: Thanks for pointing out that we could specify if women and men are part of the same couple. We explained it in 4. Discussion limitations of the study.
Reviewer #3
Comment #1: 1. Introduction
The "Introduction" section actually introduces the reader to the topic.
Author Response: Thank you for your affirmation.
Comment #2: 2. Material and methods
In my opinion, this section could be improved.
- The Actor-Partner Interdependence Model (APIM) should be better described in the text.
Author Response: Thanks for your suggestion. We’ve explained more about APIM in 2. Materials and Methods. APIM was a dyad model, which was used to study the interaction between couples. It meant that in couples, depression was not only affected by the self, but also by the spouse.
- All the questionnaires used in the work could be described in more detail way.
The author(s) and the year of creation of the scale should be mentioned. Same in cases if the validated version was used.
Author Response: Thanks for bringing up the introduction of questionnaires. We’ve added more details about measurements, including the authors and the year of creation of the scale.
Comment #3: The methods are adequate for this kind of research
Author Response: Thank you for your affirmation.
Comment #4: 4. Discussion
- In my opinion this section is too short and would be worth adding a few more literature positions.
Author Response: Thank you for suggesting adding a few more literature positions. We’ve added more discussion as well as literatures and clinical implications to better explain our findings.
Comment #5: 5. References
- References should be corrected in accordance with the instructions for the authors.
Author Response: Thanks for pointing out the references style. The references in this manuscript were in accordance with MDPI Reference List and Citations Style Guide.
Hopefully, our revision would be sufficient and satisfy the editor and reviewers. We have corrected many other errors and misspellings throughout the manuscript in colors. Thank you for your consideration again. We are looking forward to hearing from you soon.
Best regards,
Prof. Nahathai Wongpakaran, MD, FRCPsychT
Prof. Tinakon Wongpakaran, MD, FRCPsychT
Reviewer 2 Report
I would like to thank the Editor and the Authors for the opportunity to review the manuscript entitled: Factors Associated with Depression in Infertile Couples: A Study in Thailand.
I find the contribution of relevant contents for the Journal and I appreciate very much the focus. However, I think that the manuscript needs some revisions.
Could the authors specify the minimum sample size necessary for and the statistical power of the applied analyses?
Could the Authors provide the Cronbach’s alpha for each instrument, instead of stating that all measurements were greater than 0.7? Thank you.
Could the Authors indicate correlations and level of significance in the Figure representing the measurement model?
As concerns the applied statistics, could the Authors report test for interactions among the variables in the regression? It would be useful, and it could enrich the discussion that needs to be implemented. As it is now, it is mainly a report of the results without an in-depth reflection.
Could the Authors specify if women and men are part of the same couple? I do not think so; thus, this should be commented in the limitations.
I think that English is good; there may be some really minor editing issues, that could be close to typos rather than language issues.
Author Response

(The authors gave the same response as above.)

Reviewer 3 Report
Dear Authors,
Thank you for the opportunity to review the article Factors Associated with Depression in Infertile Couples: A Study in Thailand. In my opinion, the article is interesting and has potential. Some sections need to be corrected.
My comments and suggestions for authors:
1. Introduction
The "Introduction" section actually introduces the reader to the topic.
2. Material and methods
In my opinion, this section could be improved.
· The Actor-Partner Interdependence Model (APIM) should be better described in the text.
· All the questionnaires used in the work could be described in more detail way.
The author(s) and the year of creation of the scale should be mentioned. Same in cases if the validated version was used.
3. The methods are adequate for this kind of research.
4. Discussion
· In my opinion this section is too short and would be worth adding a few more literature positions.
5. References
· References should be corrected in accordance with the instructions for the authors.
Author Response

(The authors gave the same response as above.)

Round 2
Reviewer 2 Report
I thank you the Authors for replying to my previous comments. However, I am afraid the Authors misunderstood my comment as regards the lack of analysis of the couple. I did not mean the need to include fathers and mothers who were both depressed; it was a methodological issue about analyses: the study would have benefited to analyze dyads not fathers and mothers separately. Once, this misunderstanding is solved, I for sure endorse the manuscript. Thank you.
I think the English is OK
Author Response
Department of Psychiatry, Faculty of Medicine, Chiang Mai University, Chiang Mai 50200, Thailand
Dear Editor,
Re: Reviewers’ comments on manuscript ID healthcare-2422401, Factors Associated with Depression in Infertile Couples: A Study in Thailand, dated 26 June 2023
Thank you for your consideration in publishing our article. We are thankful to reviewers for their useful comments. Please see below for our point-by-point responses to the reviewers’ comments.
Reviewer #2
Comment #1: I thank you the Authors for replying to my previous comments. However, I am afraid the Authors misunderstood my comment as regards the lack of analysis of the couple. I did not mean the need to include fathers and mothers who were both depressed; it was a methodological issue about analyses: the study would have benefited to analyze dyads not fathers and mothers separately. Once, this misunderstanding is solved, I for sure endorse the manuscript. Thank you.
Author Response: Thank you for your question. I would like to clarify our analyses. For Tables 1, 2, 4, and 5, males (husbands) and females (wives) were analyzed separately (non-dyads), while in Figures 3, 4, and 5, the analysis was done in dyads- not father and mother separately.
We have added this paragraph for more clarification.
“Instead of studying fathers and mothers separately, the authors examine the interactions and dynamics between fathers and mothers as a unit or partnership in dyad analysis to understand how the two parents collaborate, communicate, and jointly influence family processes and outcomes.”
and we have deleted our previous response in the manuscript that we misunderstood it.
Hopefully, our revision would be sufficient and satisfy the editor and reviewers. Thank you for your consideration again. We are looking forward to hearing from you soon.
Best regards,
TW